# REaCH-Resiliency Engagement and Care in Health; a Befriending Intervention to Address the Psycho-Social Challenges of Vulnerable Youth in the Context of COVID-19 Pandemic: An Exploratory Trial in India

**Saju Madavanakadu Devassy** [1,2,*] 📷, **Lorane Scaria** [2] 📷, **Kalluparambil Kesavan Shaju** [1], **Natania Cheguvera** [2], **Mannooparambil K. Joseph** [1], **Anuja Maria Benny** [2] **and Binoy Joseph** [1]

1   Department of Social Work, Rajagiri College of Social Sciences (Autonomous), Cochin 683 104, India; kkshaju@rajagiri.edu (K.K.S.); emkay@rajagiri.edu (M.K.J.); binoyjoseph@rajagiri.edu (B.J.)

2   Rajagiri International Centre for Consortium Research in Social Care (ICRS), Rajagiri College of Social Sciences (Autonomous), Cochin 683 104, India; lorane@rajagiri.edu (L.S.); nataniamicheal@gmail.com (N.C.); anuja@rajagiri.edu (A.M.B.)

\*   Correspondence: saju@rajagiri.edu; Tel.: +91-989-534-6190

**Abstract:** The study explores the effectiveness, feasibility, and acceptability of a befriending intervention delivered by trained lay health workers to address the issues of the rural youth during the COVID-19 pandemic. We did an exploratory trial with 501 upskilled youth, where we randomly recruited 251 to the intervention group (REaCH) and 250 to the control group (General Enquiry Telephone Call-GETC). The outcome variables included in the study were depressive symptoms, wellbeing, and social support. The majority of the participants were females (64.2%), unmarried (63.55%), and hailed from economically poorer households (57.63%). The befriending intervention reduced depressive symptoms (OR: 0.95, $p = 0.05$) and significantly improved social support (OR: 1.03, $p = 0.000$) among participants in the intervention group. The participants in the intervention group had higher perceived social support from friends, families, and significant others when compared to the control group. Additionally, suicidality scores decreased for people in the intervention group from baseline to follow up; however, the results were not statistically significant. Befriending intervention is a practical, low-cost technique to sustain the youth in employment and ensure sustainable income. It inspires practitioners and policymakers to create mental health gatekeeping. The trial was registered prospectively on 27 July 2020 in Clinical Trial Registry India; ICMR-NIMS (Registration Number: CTRI/2020/07/026834).

**Keywords:** COVID-19; befriending intervention; rural youth; Kerala

## 1. Introduction

The COVID-19 pandemic exerts a long term impact on mental health globally, given the medical, economic, and social implications the crisis has generated. The pandemic related lockdown has become one of the most stressful events due to the unpredictability and uncertainty of the situation [1]. Health emergencies demand physical health interventions but fail to look at other social, emotional, and economic domains, a prerequisite for well-being [2]. Evidence shows that lower socioeconomic status is a predictor of common mental disorders [3–6], mainly due to limited access to social and health services [7] and lack of social support in the form of emotional, informational, and practical resources [2]. There is sufficient evidence to prove that strengthening social support by investing in available social networks, delivered through significant people in one's life, promotes social, emotional, and economic outcomes [8]. Befriending interventions, a caring relationship that regularly provides mutual support, is an effective strategy for improving the social support and well-being of vulnerable people [9–11]. According to our knowledge, there

is a paucity of research that evaluates the low-cost psychosocial interventions delivered by the non-specialised mental health workforce during this pandemic. REaCH (Resiliency Engagement and Care in Health) intervention model utilised the available human resources, using task shifting and task sharing strategies (shifting the responsibilities of professional health workers to non-medical volunteers with adequate training). There is existing evidence to show that the task sharing strategy is particularly beneficial when professional human resources are generally overburdened and outside support is urgently required [7,12].

REaCH, a "Befriending" intervention, was delivered to the youth who completed skill training, hereafter called the upskilled youth, from DDUGKY (Deen Dayal Upadhyaya Grameen Kaushalya Yojana) centres in India. DDU-GKY is an initiative of the Ministry of Rural Development (MoRD), Government of India (GOI), launched in 2014 [13]. As per the National Census 2011, 69% of India's population lives in its villages [14]. Over 55 million of the rural poor are young, between the productive ages of 15 to 35. Quality training and placement to the underserved poor rural youth can give a demographic advantage to India, especially in the context of the acute shortage of 47 million workers [15]. The project envisages that at least 70% of these trained youth should be placed based on their aptitude and skills and sustain jobs despite all odds. DDUGKY has its branches in 28 states of India. Currently, 1575 projects are being implemented by over 771 Project Implementing Agencies (PIA) across different states of India. We designed the REaCH intervention to ensure early restoration of all the youth to their jobs at the earliest opportunity. Furthermore, the unemployed and those that had lost their job were linked with the employment support service desk of PIAs to connect them with prospective jobs and employers. Additionally, the intervention also had crisis intervention and psychosocial support components.

This befriending intervention is located within the psychosocial and psycho-therapeutic framework aimed at reducing social isolation, increasing well-being, and regaining confidence by re-establishing and reengaging them with available psychosocial supports [9,16]. Furthermore, this psychosocial support model is founded on social capital and social network theories [17]. Literature on befriending identified three defining attributes: a friend-like relationship, an organised intervention package, and a negotiation of power and agency between those involved [18]. The REaCH model incorporated these components to provide a holistic and community-based support system, especially for potentially vulnerable individuals.

REaCH was developed in a four-stage process [19]. In the first stage, we conducted a literature search to identify the exposure variables and the components, the pattern of engagement, intervention design, and delivery. Some significant databases (Embase, Medline, PsycINFO, PubMed, and Google Scholar) were searched using pertinent keywords. The search could not identify any lay health worker-driven telephone befriending studies addressing the psychosocial issues of vulnerable youth that sought to enhance wellbeing and social support, especially in any prolonged crisis. In stage two, the multidisciplinary team consisted of a psychiatrist, psychologists, three psychiatric social workers, and two youth trainers to brainstorm the design, content, and delivery based on the literature and professional experience. In stage three, we conducted a state-wide audit to identify and prioritise the needs of the youth [20]. We did the state-wide audit with 14,430 rural youth trained through DDU-GKY centres across Kerala state. This study informed three areas of concern: economic and job-related issues, health and various life stressors, and social and psychological distress. The data from the rapid review, information gathered through the state-wide audit, discussion, and consultations with the experts informed various domains of intervention, including the targeted audience, intervention components, duration, and frequency of intervention.

India faces significant unemployment rates, especially among youth, across the years, which marked 23.01% in 2019 [21]. The situation is further worsened by the shrinking economic growth imposed by COVID-19 in India [22]. In a country like India, where most

of the population is under 25, the pandemic and associated impacts leave a considerable burden, including loss of jobs and delayed economic independence [23]. To address the economic vulnerability of youth in India, we proposed the Resiliency Engagement and Care in Health (REaCH) intervention to increase social support from family and friends, improve support networks, and reduce depressive symptoms. In this paper, we outlined the results of a pilot exploratory trial conducted on 439 upskilled youth from the DDUGKY centre of Rajagiri College of Social Sciences, an autonomous college in Kerala, during the time of COVID -19 lockdown.

## 2. Materials and Methods

### 2.1. Study Design

We conducted an exploratory trial of REaCH intervention between 29 July and 10 September 2020. The trial randomised participants to REaCH intervention or GETC group based on computer-generated random numbers. There were two data collection points: before the intervention and one month after. The trained and supervised DDUGKY staff made three structured phone calls to all the participants in the REaCH arm, while the untrained staff made GET calls to participants in the control arm.

### 2.2. Ethics

After receiving their oral consent, we enrolled the participants in the intervention phase, explaining the intervention objectives and a brief description of the purpose, content, implications, and risks of participation. This oral consent was audio recorded with their permission. Ethics committee approval for the exploratory trial was obtained from the Rajagiri Institutional Review Board (IRB) of Rajagiri College of Social Sciences (Reference Number: RIRB 2004). The trial was registered prospectively on 27 July 2020 in Clinical Trial Registry India; ICMR-NIMS (available at http://ctri.nic.in/Clinicaltrials/showallp.php?mid1=459 53&EncHid=&userName=CTRI/2020/07/026834, accessed on 10 November 2021).

### 2.3. Study Participants

We recruited the participants from a single DDUGKY centre. We used broad inclusion criteria in the participant recruitment, i.e., the upskilled youth who were currently working or not working or searching for suitable jobs. Out of 1036 potential participants, we excluded 535 for not meeting the inclusion criteria, lack of consent, or not being accessible to participate in the study. Out of 501, we randomised 251 to the REaCH and 250 to the three GET calls cohorts (See Figure 1). Out of the randomised participants, 62 didn't respond to the telephonic call and was further excluded. Finally, 439 participants (251 in the intervention group and 188 in the control group) were allocated and included in the analysis.

### 2.4. Randomisation

We randomly assigned the participants in a 1:1 ratio via a computer-generated random number list. We allotted the odd numbers to the intervention arm and even numbers to the control arm. The trial team, trial manager, and staff members were blinded to the allocation codes during the trial. A computer technician did the recruitment. The randomisation list was password protected and had not been shared with anyone involved in the study. The use of computer-based data allocation helped in masking the outcomes from the intervention providers. The group allocation of the participants was masked by introducing general enquiry telephone calls with the control group. To eliminate contamination, we separated the structured intervention team and general enquiry team physically and concealed the group to which they belonged and the type of instruction they received. Lockdown related social distancing and work from the home mode of functioning of the staff made this masking easy. A committee of professionals and a team of researchers provided additional oversight to the trial.

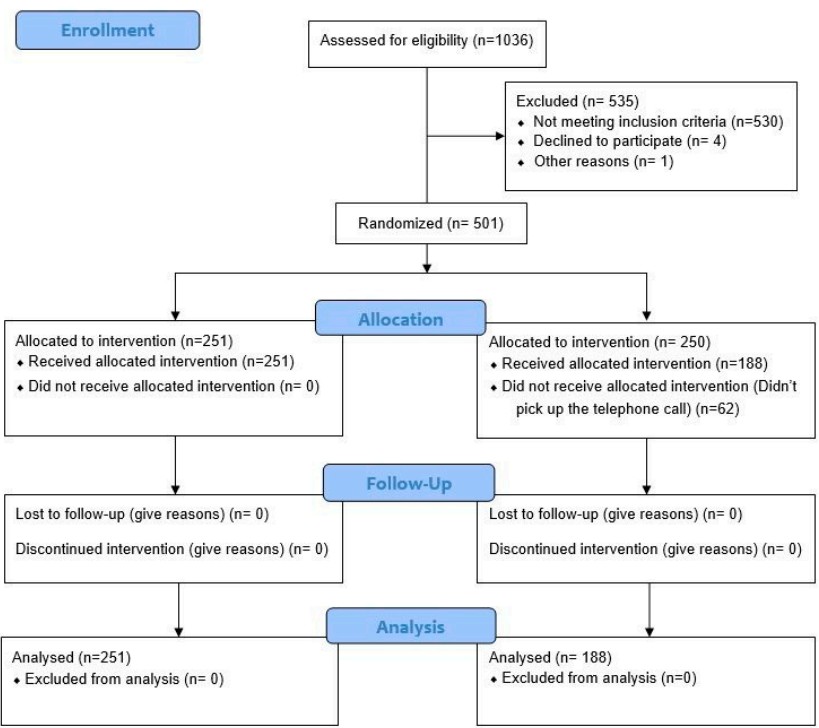

**Figure 1.** Participant Recruitment (*n* = number of respondents).

*2.5. Intervention*

The finalised REaCH intervention package had three components designed in a non-linear, iterative fashion keeping a user-centric perspective to prototype it to test its efficacy, feasibility, and acceptability. The critical areas covered in the package are

1. proactive engagement and crisis intervention,
2. problem-solving oriented supportive therapy, and
3. assertive linkage with community resources.

The program was designed in such a way as to be able to customise it based on individual differences of the persons. The assessments and the modules were delivered over the phone. The development process of REaCH and the protocol for its testing have been described in detail elsewhere [19].

*2.6. Training of the Staff*

The intervention team consisted of DDUGKY Project staff members, who completed at least one year. Prelude to the intervention, the REaCH intervention team received one day (6 h) training on content and process of intervention. In addition to online training, we provided them with an intervention manual, video of the training material, audio clips of model interviews, and a module on frequently asked questions to the REaCH intervention team. The intervention manual has guidelines on developing relationships with clients, introduction and orientation to befriending, strategies to manage participant distress, confidentiality aspects, and safety issues for both staff and participants. We introduced two-layer supervision to ensure fidelity to the protocol and to facilitate referral in case of mental health emergencies. Two non-medical mental health professionals (a psychiatric social worker and a clinical psychologist) supervised the staff in the first level. In the second level, the supervising team consisted of a psychiatrist, two psychiatric social workers, and two clinical psychologists.

*2.7. Intervention Arm: Resiliency Engagement and Care in Health (REaCH)*

We used a semi-structured intervention manual to allow sufficient flexibility to suit the needs of each participant. REaCH intervention consisted of three modules. In module 1,

we assessed participants for various psycho-somatic and social indicators of distress such as sleep, appetite, interpersonal relationships, adjustments, and work-life to determine the level of disturbance. The first level of the intervention included proactive engagement and crisis intervention focused on psycho-education, self-absorbing activity engagements, and symptom-based intervention. Module 2 consisted of brief problem-solving support-oriented therapy with a specific focus on their current felt needs and problems. We targeted prioritised needs through the mobilisation of both formal and informal resources. Module 3 focused on assertive linkage with employment opportunities and community resources, encouraging help-seeking behaviours and educating about the protective strategies. These three-module intervention packages were unpacked in three sessions of 30 min to 1-h duration. Three structured phone calls were made to the participants, one week apart, for one month.

### 2.8. Control Arm: General Enquiry Phone Calls (GETC)

Participants randomised to the control arm were not given any intervention. Instead, they received three general enquiry phone calls lasting 5 to 30 min. The general inquiry dealt with physical health and precautions that need to be taken to protect themselves from the pandemic and how they cope with lockdown-related issues. The main focus of the phone call was on psycho-education based inquiries on COVID-19. No specific training was given for the staff involved in GET calls in the control arm.

### 2.9. Study Tools

#### 2.9.1. Assessments and Procedures

After accepting the informed consent sheet, the participants were auto-directed sequentially to the demographic information questionnaire and other standardised tools. The questionnaire included the Patient Health Questionnaire (PHQ-9) [24], the Multidimensional Scale of Perceived Social Support (MSPSS) [25], and the WHO-Wellbeing index 5 [26], in addition to the sociodemographic details. These questionnaires were translated into the local languages and reverse-translated for accuracy. Two bilingual translators, one familiar with the concepts under consideration and another a language expert, translated the questionnaire independently in the adaptation process. Later, they synthesised their versions, addressed issues, and decided on appropriate words and phrases to communicate the concepts. This version was then back-translated to the original language, and later, the expert committee consolidated and ensured the conceptual and experiential equivalence. The non-threatening nature of the variables and web based, self-administered, pre post-assessment questionnaires minimised the effects of social desirability bias. Moreover, the longitudinal nature of the study also addressed the common method bias. We also provided detailed instructions and sufficient explanations on the face page of the online survey.

#### 2.9.2. Outcome Measures

The demographic variables were age, gender, marital status, occupation, education level and colour of ration card. We measured the presence of depressive symptoms using the PHQ-9, with scores of 1–4, 5–9, 10–14, 15–19, 20–27, indicating minimal, mild, moderate, severe, and extreme depressive symptoms. Well-being was measured using WHO-5. The total row score ranging from 0 to 25 is multiplied by 4 to provide the final score. In this range, 0 represents the worst possible wellbeing, and 100 represents the best possible wellbeing. The MSPSS measured perceived social support (SS) from three sources: family, friends, and a significant other. This measure contains 12 questions which were rated on a 7-point scale as "Very Strongly disagree", "Strongly disagree", "Mildly disagree", "Neutral", "Mildly agree", "Strongly agree", "Very strongly agree". The MSPSS has high internal consistency and moderate construct validity as the SS scores were negatively correlated to anxiety ($r = -0.18$; $p < 0.01$) and depression scores [27]. All the measures used were cross-culturally validated for sensitivity and reliability [28–31]. Cronbach's alpha

coefficient for WHO-5 items, MSPSS, and PHQ-9 were 0.85 [32], 0.70 [27], and 0.89 [24] respectively. We conducted a post-intervention follow-up assessment through an online platform link for both the control and intervention arm using the same baseline survey assessment tools.

*2.10. Data Analysis*

Statistical tests used a *p*-value less than 0.05 for significance. All statistical analysis procedures were done using STATA 14, R version 3.6.3 and SPSS version 24. We calculated baseline summary statistics (mean, standard deviation, percentage) based on groups. Chi-square tests and T-tests were performed to test the significance between the study variables. An odds ratio of the outcome variables for the post-assessment was calculated using logistic regression modelling, and 95% confidence intervals were presented.

**3. Results**

The study included 439 participants. At baseline, the mean age of the participants was 25 years (S. D = 5.7). The majority of the participants were females (64.2%), unmarried (63.55%), and hailed from economically poorer households (57.63%). In addition, 42.8% of the respondents were either unemployed or homemakers even though most of the participants had completed minimum education to be employed (91.2%) (Table 1). Intervention and general enquiry call groups were similar in terms of gender, occupation, education, and marital status. At baseline, 8.2% of people were reported to have moderate and above levels of depressive symptoms, 55% were females, and 80.6% were below 30.

**Table 1.** Participant characteristics of intervention and control groups at baseline.

| Variables | Total | General enquiry Telephonic Calls Group (*n* = 188) | REaCH Befriending Intervention Group (*n* = 251) |
|---|---|---|---|
| Age, in years | 25.1 (5.7) | 25.5 (6.2) | 24.8 (5.2) |
| Gender | | | |
| Female | 282 (64.24%) | 122 (64.89%) | 160 (63.75%) |
| Male | 157 (35.76%) | 66 (35.11%) | 91 (36.25%) |
| Education | | | |
| Completed 10th | 38 (8.66%) | 19 (10.11%) | 19 (7.57%) |
| Completed 12th | 174 (39.64%) | 77 (40.96%) | 97 (38.65%) |
| Completed graduation | 206 (46.92%) | 81 (43.09%) | 125 (49.80%) |
| Completed post-graduation | 21 (4.78%) | 11 (5.85%) | 10 (3.98%) |
| Occupation | | | |
| Paid job | 147 (33.49%) | 63 (33.51%) | 84 (33.47%) |
| Self employed | 5 (1.14%) | 4 (2.13%) | 1 (0.40%) |
| Housewife | 54 (12.30%) | 26 (13.83%) | 28 (11.16%) |
| Unemployed | 134 (30.52%) | 48 (25.53%) | 86 (34.26%) |
| Daily wage | 99 (22.55%) | 47 (25.00%) | 52 (20.72%) |
| Marital status | | | |
| Unmarried | 279 (63.55%) | 120 (63.83%) | 159 (63.35%) |
| Married | 153 (34.85%) | 67 (35.64%) | 86 (34.26%) |
| Divorced/widowed/separated | 7 (1.59%) | 1 (0.53%) | 6 (2.39%) |
| Colour of ration card [a] | | | |
| Yellow | 35 (7.97%) | 15 (7.98%) | 20 (7.97%) |
| Pink | 218 (49.66%) | 95 (50.53%) | 123 (49.00%) |
| Blue | 147 (33.49%) | 61 (32.45%) | 86 (34.26%) |
| White | 39 (8.88%) | 17 (9.04%) | 22 (8.76%) |

[a] Yellow and Pink indicate below poverty line; Blue and White indicate above poverty line.

Mean scores of wellbeing, depressive symptoms, and social support for the intervention group at baseline were 15.8, 4.8, and 65.4, which changed to 17.4, 4.5, and 67.9 respectively after the befriending intervention (Table 2). Mean scores of the subscales of social support were significant others (21.8 ± 4.8), family (22.2 ± 4.8), and

friends (20.9 ± 5.01), respectively. Paired t test showed significant results for wellbeing (t (250) = −3.99, *p* = 0.000) and social support (t (250) = −2.55, *p* = 0.005) between baseline and follow up for the intervention group. Further, logistic regression revealed that social support (or: 1.03, *p* = 0.000) was significantly higher and depressive symptoms (OR: 0.95, *p* = 0.055) were lower for participants in the REaCH intervention compared to the general enquiry group. In addition, suicidality and levels of depressive symptoms were lower, and well-being was higher for the intervention group; however, the results were not statistically significant.

**Table 2.** Outcome Measures: wellbeing, depressive symptoms, and social support between Befriending Intervention group (REaCH) and general enquiry telephonic calls (GETC).

| | REaCH—Befriending Intervention | | | GETC | | | REaCH Model v/s GETC | |
| --- | --- | --- | --- | --- | --- | --- | --- | --- |
| | Mean (S.D) | T(df) | *p* | Mean (S.D) | T(df) | *p* | OR (95% CI) | *p* |
| Wellbeing | | | | | | | | |
| Baseline | 15.82 (5.7) | −3.99 (250) | 0.000 | 15.26 (5.2) | −3.12 (187) | 0.010 | 1.02 (0.98, 1.05) | 0.274 |
| Post-assessment | 17.42 (4.9) | | | 16.87(5.5) | | | | |
| Depressive symptoms | | | | | | | | |
| Baseline | 4.80 (3.5) | 0.88 (250) | 0.812 | 4.93 (3.5) | −1.06 (187) | 0.143 | 0.95 (0.90, 1.00) | 0.055 |
| Post-assessment | 4.53 (3.6) | | | 5.26 (4.1) | | | | |
| Social Support | | | | | | | | |
| Baseline | 65.37 (13.8) | −2.55 (250) | 0.005 | 64.5 (12.8) | 1.96 (187) | 0.97 | | |
| Post-assessment | 67.89 (13.4) | | | 62.84 (14.0) | | | 1.03 (1.01, 1.04) | 0.000 |
| Suicidality | | | | | | | | |
| Baseline | 0.25 (0.6) | 0.14 (250) | 0.55 | 0.24 (0.6) | −1.23 (187) | 0.10 | 0.80 (0.59, 1.08) | 0.156 |
| Post Assessment | 0.24 (0.6) | | | 0.32 (0.6) | | | | |

Table 3 represents the odds ratio, 95% confidence intervals, and *p* values from logistic regression models, along with the difference in mean values of the subscales of social support. Regression analysis revealed that REaCH intervention was associated with notable changes in support from significant others (OR: 1.04, *p* = 0.013), support from family (OR: 1.04, *p* = 0.012), and support from friends (OR: 1.04, *p* = 0.004).

**Table 3.** Social support from significant others, family, and friends—odds ratio, 95% CI of odds ratio, and *p* values.

| | REaCH—Befriending Intervention | GETC | REaCH Model v/s GETC | |
| --- | --- | --- | --- | --- |
| | Mean (S.D) | Mean (S.D) | Odds Ratio (95% CI) | *p* |
| Social support-Family | | | | |
| Baseline | 22.26 (5.1) | 22.11 (4.4) | | |
| Post-assessment | 21.71 (5.6) | 20.20 (6.7) | 1.04 (1.00, 1.07) | 0.012 |
| Social Support-friends | | | | |
| Baseline | 21.24 (4.9) | 20.66 (5) | | |
| Post-assessment | 20.84 (5.4) | 19.14 (6.8) | 1.04 (1.01, 1.08) | 0.004 |
| Social support-Significant others | | | | |
| Baseline | 21.85 (5.1) | 21.76 (4.6) | | |
| Post-assessment | 20.52 (5.3) | 20.06 (6.9) | 1.04 (1.00, 1.07) | 0.013 |

## 4. Discussion

This paper explored the feasibility, acceptability, and effectiveness of a structured befriending intervention compared to general enquiry call intervention by the DDUGKY staff members to reduce levels of depressive symptoms and enhance wellbeing and social support. This study suggests that REaCH is potentially a safe and engaging intervention that non-medical mental health workers can deliver to many people. It is evident from the results that the REaCH has improved social support but is unsure about its contribution to the wellbeing and moderate depressive symptoms. Conclusive insights into why

this intervention has not positively impacted on depressive symptoms wellbeing are not established. The possible explanation could be that the interplay of different individual, family, and society-related factors, especially economic vulnerability, could have nullified the effect on wellbeing and depressive symptoms. In addition to the direct impact of financial stress on mental health, it can also offset the protective factors of social support, explaining the lack of positive results in wellbeing and depressive symptoms. It is also probable that negative cognitive patterns associated with the economic crisis in times of the pandemic situation resulted in underestimating social support's positive potentials, consequently resulting in hopelessness. Job loss and uncertainty about future job opportunities could have had complex interactive effects on depressive symptoms and well-being, which further studies need to address.

Nevertheless, though not significant, the improvement can instil hope and protect the vulnerable youth from adverse mental health outcomes, even suicidality. For example, the reduction in the mean score of suicidality in REaCH compared to GETC suggests that the intervention triggered the change process positively but suggests alterations in the design. Thus, this study's findings can inform the final design of the package of care, especially the variables of duration and time factor, which can further be tested using randomised control trial. For instance, study findings suggest longer follow up periods, as the duration of one month was insufficient to improve well-being, mainly when the socioeconomic vulnerability context remained unfavourable due to lockdown.

To the best of our knowledge, no previously published studies employed telephone befriending to manage psycho socioeconomic challenges in vulnerable youth with a national focus. This feasibility study has informed the preliminary evidence regarding the structure, design, feasibility, acceptability, and efficacy. Additionally, it proved that the health crisis neutralised the effects of the strategies that proved to be effective in normal situations. For instance, two previous meta-analyses showed that the befriending intervention reduced depressive symptoms, emotional distress, and social support [16], improved quality of life, and decreased loneliness [33], which partially proved true in our study. However, our findings are comparable to previous studies on befriending-based interventions that enhanced social support [9–11].

Further subgroup analysis on social support revealed that, of the three sources, the lowest perceived support was reported from significant others, followed by family and the highest from friends. This finding does not agree with the conclusions from Ravindran and Myers [34], who found that Indians are likely to list family and extended family as the primary sources of support. The possible reason could be that the strict lockdown and long-term home isolation would have led to increased levels of family conflicts and structural and other situational stressors [35–37].

Additionally, due to prevailing collectivist norms in Indian societies, only the family's support directly impacts the relationship between mental health and social support [38]. The cultural norms prevalent in India typically do not support sharing concerns beyond their immediate family network or inner circle of the social network, which in most cases comes from a spouse or a relative [39]. Thus the social support from significant others and friends may not have the potential to reduce depressive symptoms and enhance well-being in normal situations. However, the reverse is true in the context of the pandemic.

The current study's findings may imply that social support can positively impact depressive symptoms only when insulated from economic stressors. Economic stress could directly affect depression symptoms and negatively influence an essential protective factor, social support. The study found that the befriending intervention was helpful to link people with social resources but not economic resources. Therefore, it calls for a strong focus on the economic domain, especially when the crisis interventions are designed for vulnerable youth.

Regarding the human resource in service delivery, the task-shifting and task-sharing between lay personnel are already proven to improve access to health care services and mental health [40]. In addition, the use of lay mental health workers in the care of subjects

with common mental disorders was found cost-effective and cost-saving [41]. Tapping the untapped non-professional mental health human resource of DDUGKY staff resulted in reduced cost, improved sustainability, and increased acceptability. In addition, the additional mental health training these staff members have received as part of this intervention would make them better informed about the mental health needs of the youth joining the DDUGKY programs. The additional component of the REaCH befriending intervention compared to GETC was a structured intervention with well-defined components, delivered in an emotionally connected friend-like relationship without compromising the negotiation of power. REaCH model is an innovative strategy to address the barriers to maintaining their jobs, thereby achieving the government program's goals. The insights would inspire the government policies to incorporate similar strategies. They would encourage other PIAs and non-government implementing agencies to design innovative, cost-effective strategies to ensure the sustainability of all the government-funded programs in India, which are otherwise fragmented, incomplete, or inconsistent [42].

*Strengths and Limitations of the Study*

For the first time, the befriending intervention was modelled with four specific components: proactive engagement, crisis intervention, brief problem-solving support-oriented therapy, and assertive linkages. To the best of our knowledge, limited studies employed structured telephone befriending intervention to manage psychological and social determinants of vulnerable groups, particularly for the youth from low-income families. Another strength is that we have zero loss in follow up in the REaCH group. Volunteerism and task-shifting strategies made this intervention most cost-effective. The only cost incurred for the intervention was the telephone and internet charges, for which the amount was negligible. The intervention manual and frequently asked questions were e-content, so printing and stationery costs were also minimal. The impact of REaCH intervention has to be looked at from the changes in the control group scores. The control group had a significant increase in adverse outcomes and reduced positive effects such as social support and wellbeing. Our study had its limitations as well. Identifying the possible factors other than social support contributing to mental health is a question unaddressed in this study. It is crucial to understand why social support could not significantly reduce depressive symptoms and increase wellbeing. The potential effect of economic issues in offsetting the positive impact of social support in reducing depressive symptoms and increasing wellbeing needs to be studied in detail. A smaller sample size and short interval between baseline and follow up might also be a major limitation of the study. It is important to note that the study's findings have opened new avenues to future research involving larger samples and modified packages of care with a more specific focus on economic and occupational supports than on social support to improve the outcomes. The study results also point out that the duration and length of intervention need to be revisited to affect outcome variables better.

## 5. Conclusions

Though the mechanisms linking economic stress and depressive symptoms and wellbeing are not well understood from this study, the study results suggest an additional focus in the intervention: financial and occupational supports. In the global mental health context, the findings strengthen the case for adopting the structured befriending intervention through the nationwide network of DDUGKY centres as a mental health promotion strategy. Replicating the intervention content and process across India would benefit more than 1.2 million upskilled youth from different Program Implementation Agencies under this scheme. Though the intervention was individual-focused, in the collectivist context of India, intervention impact extends to their families and even to their extended families. REaCH extends emotional support at the individual level, educates on strategies to alleviate the crisis, and offers practical help and opportunities (e.g., resuming jobs). At the policy level, we aim to significantly affect new programmatic policy and clinical guidelines to

improve economically vulnerable groups' psychological and social needs. Ultimately, we aim to make social and health services accessible and affordable to all, consequently improving social support systems and the overall wellbeing of the upskilled youth.

**Author Contributions:** Conceptualization, S.M.D.; methodology, S.M.D.; formal analysis, L.S.; investigation, L.S., A.M.B. and N.C.; writing—original draft preparation, S.M.D.; writing—review and editing, L.S., K.K.S., N.C., M.K.J., A.M.B. and B.J.; supervision, S.M.D. and B.J.; project administration, S.M.D.; funding acquisition, S.M.D. All authors have read and agreed to the published version of the manuscript.

**Funding:** This research was funded by Rajagiri College of Social Sciences (Autonomous) under the grant number RCSS/01/2020. Funders have no role in the design of the study and collection, analysis, and interpretation of data and in writing the manuscript.

**Institutional Review Board Statement:** The study was conducted according to the guidelines of the Declaration of Helsinki, and approved by the Institutional Review Board Sciences (Reference Number: RIRB 2004) of Rajagiri College of Social Sciences (Kalamassery) on 20 May 2020.

**Informed Consent Statement:** Informed consent was obtained from all subjects involved in the study.

**Data Availability Statement:** The datasets generated and/or analysed during the current study are available from the corresponding author on reasonable request.

**Acknowledgments:** The authors wish to thank all the students of the DDUGKY programme of Rajagiri College of Social Sciences (Autonomous), Cochin, Kerala, India who were part of the data collection. Authors also express their gratitude to all the DDUGKY staff who were part of the project in collecting the data and delivering the telephonic intervention. Authors would like to specifically mention Rajeev SR, Coordinator, DDUGKY centre of Rajagiri College of Social Sciences (Autonomous), Cochin, Kerala, India for his special effort in coordinating and managing the project within the centre.

**Conflicts of Interest:** The authors declare no conflict of interest. The funders had no role in the design of the study; in the collection, analyses, or interpretation of data; in the writing of the manuscript, or in the decision to publish the results.

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
