# Peer review of "REaCH-Resiliency Engagement and Care in Health; a Befriending Intervention to Address the Psycho-Social Challenges of Vulnerable Youth in the Context of COVID-19 Pandemic: An Exploratory Trial in India"

_sustainability, doi:10.3390/su132212920_

Round 1

Reviewer 1 Report

Introduction:

  1. You intruded on several studies that are related to your topic, but the serious problem is that there is no theory to support the variables in your theoretical framework.
  2. I agree that the COVID-19 in India is a topic worth studying, but you should explain why you must survey it in India instead of other countries.

Methods:

  1. Because there is no theory to support your theoretical model, the data should be a longitudinal section to confirm the causal relationship in your theoretical model.
  2. How do you handle the common method bias and social-desirability bias?

Discussion

  1. You only intruded on the analysis results and cited serval research. I suggest you should rewrite the discussion to demonstrate the contributions.

Reviewer 2 Report

The article presents an interesting introductory report on the possibility of using the economic method of psychological support for adolescents in crisis situations, such as the COVID-19 pandemic, which may be related to an increase in the sense of social support. A sense of social support is known to play a very important role in coping with difficulties, which is especially important among young people, who have to deal with many problems of everyday life, exacerbated by the pandemic.

Noteworthy is a satisfactory, fairly detailed description of the individual elements of the intervention preparation, including a clear presentation of the selection of study groups and their socio-demographic characteristics.

However, the manuscript also has several weaknesses that should be corrected or discussed prior to publication. Thinking Notes are included below:

  1. The title contains words starting with small and capital letters - the way of selecting words beginning with a capital letter is incomprehensible.
  2. In my opinion, writing in an article about measuring „depression” and determining the level of depression in the respondents is incorrect. No questionnaire entitles to such a diagnosis. It would be better to use the phrase e.g. „severity of depression symptoms”.
  3. I propose to replace the word „happenings” in line 43 with another, more appropriate one.
  4. Some abbreviations such as ReACH in line 60 or DDU-CKY in line 70 are explained later, while for the sake of clarity it would be better to explain them at the first indication in the text.
  5. Check the article for correctness of punctuation. Words beginning with a capital letter appear in the text, which looks like to be an editorial error, for example, 'Upskilled youth' on line 70 or 'Indicates' on line 234.
  6. On line 102 the authors indicate "DDUGKY Staff made three structured phone calls to all the participants in the REaCH arm" while on line 178 they indicate "These three-module intervention packages were unpacked in four sessions ...". So, how many conversations with the participants were there?
  7. The REaCH description did not mention the intervals between the interviews with the participants. Was it indicated in the instructions?
  8. The description of the questionnaires used indicated: "These questionnaires were translated into the local languages ​​and reverse-translated for accuracy" - were the adaptation procedures performed by the authors? Were the basic psychometric properties of the tools prepared in this way examined?
  9. What are the "SS scores" in line 207?
  10. Various spellings for the name REaCH appear in the text. For example, in line 218 „reach” and in line 244 „REACH”. It would be worth standardizing the record. 

    I hope my comments will help to improve the quality of the manuscript.

Reviewer 3 Report

Overall this is important and valuable work.  Your passion and care for the youth is obvious.

My biggest concern is that this paper is written in a way that expects the reader to be already familiar with the issues, study and settings.  The authors are too close to the study and need to revise the paper with the assumption that the audience has no knowledge or background in this topic. 

Line 54, This statement should be moved to the methods section.  Restate this and avoid using 2nd person (we/us/our) in the introduction.  Try “A database search  revealed a lack of studies focused on ……”

Line 67 Need to specify what you mean by “this specific population”

Intro overall is very short and needs more content – Why youth?  how are they more vulnerable? What is your organization? Who is doing the GETC and what is it? What is “upskilled”?  What is the Anonymous DDU-GKY center?   Incorporate the content from 2.1 overview into the Introduction section to help your reader understand the setting and background before introducing your hypothesis and starting into Methods. 

Were the GET call already happening and part of your regular practices?  Were this added only for this research project?  This needs to be clear.  If GET calls were not routine, then the GET calls are a 2nd intervention and this study should have included a true control group.  In line 264 you call the GET call and intervention – in 337 you call them the control group - the distinction here are very important and must be clarified.

Line 103 – add number of GET calls made

Line 136 – clarify who you mean by “their”

2.6 Intervention – I think a lot of this content should be move to the introduction This is prior work.  The methods should focus on the new portion of this larger project.

Line 261 – remove test

Line 283 replace “them” with “vulnerable youth”

Line 309 replace “could” with “may”

Line 311 replace “could” with “may”

Line 345 – suggest that sample size could have been a limitation as your are looking for relatively small changes in outcome and several were nearing but not quite statistically significant – and mention the short interval between data collection – repeating data collection in another 30 days may have strengthen your findings.

Besides, please try to improve English.

Round 2

Reviewer 1 Report

Thank you for revising the article according to my comments. After reviewing your article again, I found that the variables and hypotheses of this article have been empirically examined in serval studies, so its contribution should not be enough for publication. Also, you should propose why this research has made a higher contribution because there is no single theory to fully support your theoretical model). Besides, the plagiarism rate of this manuscript is still high.

Reviewer 2 Report

The authors made most of the suggested corrections, diligently completing the manuscript. It is a pity that the authors did not indicate the basic properties of the tests used in their own research sample (e.g. Cronbach's alpha coefficients). In my opinion, the manuscript can be published.

Reviewer 3 Report

This paper is much easier to read and follow now.

Author Response

Thank you for the support and continued interest in our paper.